# Perspectives of Targeting Autophagy as an Adjuvant to Anti-PD-1/PD-L1 Therapy for Colorectal Cancer Treatment

**DOI:** 10.3390/cells14100745

**Published:** 2025-05-20

**Authors:** Nasrah ALKhemeiri, Sahar Eljack, Maha Mohamed Saber-Ayad

**Affiliations:** 1College of Graduate Studies, University of Sharjah, Sharjah P.O. Box 27272, United Arab Emirates; 2Sharjah Institute for Medical Research, University of Sharjah, Sharjah P.O. Box 27272, United Arab Emirates; sfadlalla@sharjah.ac.ae; 3Department of Pharmaceutics, Faculty of Pharmacy, University of Gezira, Wad Madani 21111, Sudan; 4Department of Clinical Sciences, College of Medicine, University of Sharjah, Sharjah P.O. Box 27272, United Arab Emirates; 5Department of Pharmacology, Faculty of Medicine, Cairo University, Cairo 12211, Egypt

**Keywords:** autophagy, colorectal cancer, immunotherapy, tumor immune microenvironment, anti-PD-1/PD-L1, microsatellite instability, immune checkpoint inhibitors, therapeutic resistance

## Abstract

Colorectal cancer (CRC) is the third most common cancer in the world, with increasing incidence and mortality rates. Standard conventional treatments for CRC are surgery, chemotherapy, and radiotherapy. Recently, immunotherapy has been introduced as a promising alternative to CRC treatment that utilizes patients’ immune system to combat cancer cells. The beneficial effect of immune checkpoint inhibitors, specifically anti-PD-1/ PD-L1, has been ascribed to the abundance of DNA replication errors that result in the formation of neoantigens. Such neoantigens serve as distinct flags that amplify the immune response when checkpoint inhibitors (ICIs) are administered. DNA replication errors in CRC patients are expressed as two statuses: the first is the deficient mismatch repair (MSI-H/dMMR) with a higher overall immune response and survival rate than the second status of patients with proficient mismatch repair (MSS/pMMR). There is a limitation to using anti-PD-1/PD-L1 as it is only confined to MSI-H/dMMR, where there is an abundance of T-cell inhibitory ligands (PD-L1). This calls for investigating new therapeutic interventions to widen the scope of ICIs’ role in the treatment of CRC. Autophagy modulation provides a good example. Autophagy is a cellular process that plays a crucial role in maintaining cellular homeostasis and has been studied for its impact on tumor development, progression, and response to treatment. In this review, we aim to highlight autophagy as a potential determinant in tumor immune response and to study the impact of autophagy on the tumor immune microenvironment. Moreover, we aim to investigate the value of a combination of anti-PD-1/PD-L1 agents with autophagy modulators as an adjuvant therapeutic approach for CRC treatment.

## 1. Introduction

Colorectal cancer (CRC) is the third most prevalent cancer type [1,2]. It currently accounts for 9.6% of all malignant tumors and is the most prevalent type of cancer in the digestive system [2,3]. Its current incidence and mortality in men are 10.4% and 9.2%, and in women, 8.9% and 9.4%, respectively [2]. In the upcoming years, it is anticipated to surpass the overall mortality rate from heart diseases [4]. CRC is more prevalent in the age group from 65 to 74 [5]. Nonetheless, over the past few years, several reports have been published that highlight a concerning trend—there has been a noticeable increase in cases of colorectal cancer among younger adult patients, specifically those under the age of 50 [2,6].

The treatment of colorectal cancer depends on various factors, including the patient’s general health and the tumor’s type, stage, and location. Surgical excision is the conventional and primary treatment for localized resectable tumors, which aims to eliminate any adjacent lymph nodes. In case of non-resectable tumors, and to eradicate any remaining cancer cells and reduce the risk of recurrence, adjuvant therapies such as chemotherapy, radiation therapy, small molecule inhibitors/targeted therapy, and recently, immunotherapy are used alone or in combination [7]. Despite the marked improvement in surgical techniques and the development of chemotherapeutic and targeted agents, CRC remains a devastating affliction worldwide, especially when it metastasizes [8]. The 5-year relative survival rate for CRC patients varies according to stage and location. For instance, it is 91% for the localized stage, it drops to 73% when the tumor extends to adjacent tissues or regional lymph nodes, and reaches just 14% when the cancer has metastasized to distant parts of the body [6].

Immunotherapy, including immune checkpoint inhibitors (ICIs) such as programmed cell death protein-l/programmed death-ligand 1 (anti-PD-1/PD-L1), has emerged as a promising and relatively safe therapeutic option for different types of cancers, including CRC, melanoma, and non-small cell lung cancer [9]. This review will focus on the potential role of autophagy modulation in enhancing the effect and widening the scope of anti-PD-1/PD-L1 drugs for CRC treatment. First, we will illustrate clinically approved anti-PD-1/PD-L1 therapies as a promising treatment option for immunotherapy-sensitive CRC and microsatellite instable CRC (MSI-H/dMMR) and briefly discuss their resistance development to anti-PD-1/PDL-1. Secondly, we will introduce the role of autophagy by unfolding its various functions in the CRC tumor microenvironment (TME), immunity, and regulation of the PD-1/PD-L1 immune checkpoint pathway. Then, we will feature the available preclinical models for testing immunotherapies and the optimum in vivo models to monitor autophagy, emphasizing the consideration for better simulation of the clinical setting. Finally, the therapeutic perspective of targeting autophagy will be highlighted.

## 2. Overview of PD-1/PD-L1

Immune checkpoint inhibitors modulate the interaction between immune cells, including CD4+ and CD8+ T-cells, B-cells, macrophages, dendritic cells (DCs), tumor-infiltrating lymphocytes (TILs), and tumor cells [10,11]. By releasing brakes that tumors use to blunt protective anti-cancer responses, ICIs unleash T-cells to attack a subset of tumors that express specific “immune checkpoints” [12]. PD-1 and its ligand PD-L1 have shown the most promising treatment outcomes to date and were approved by the FDA for the treatment of MSI-H/dMMR CRC [13]. The PD-1 or CD279 belongs to the immunoglobulin gene superfamily; it is a type of transmembrane glycoprotein [14]. The binding between PD-1 and its ligands PD-L1 (CD274; B7-H1) and PD-L2 (CD273; B7-DC) conveys inhibitory signals to impede the activity of T effector cells [15,16]. Many cancer types exhibit overexpression of PD-1, resulting in continuous binding of PD-1 (on T-cells) to its ligand, PD-L1 (on cancer cells). Consequently, there is a constant high level of PD-1/PD-L1 signals, leading to the suppression of T-cell activation and the development of antigenic tolerance. This, in turn, enables cancer cells to evade the immune system and support a high tumor proliferation rate [17].

### 2.1. PD-1/PD-L1 in CRC

Several studies and clinical trials have indicated that microsatellite stability level and mismatch repair (MMR) status may help predict which CRC patients may benefit most from immunotherapy [18]. Patients with the microsatellite instability-high and deficient mismatch repair (MSI-H/dMMR) phenotype, characterized by high tumor mutation burden (TMB), tend to have better survival rates than those with the microsatellite stability and proficient mismatch repair (MSS/pMMR) phenotype [19]. Specifically, individuals with MSI-H/dMMR colorectal cancer are often responsive to anti-PD-1/PD-L1 immunotherapy, making it a promising treatment option for this subgroup of patients, as it is currently approved for this group only [13]. Conversely, the MSS/pMMR group, which accounts for approximately 85% of colorectal cancers, is more prevalent in the earlier stages of cancer (stage I and stage II) and is associated with a higher risk of cancer recurrence notably, 95% of the MSS/pMMR tumors eventually metastasize [20]. Personalizing treatment selection based on tumor genetics represents an avenue to improve outcomes [21].

Several clinical trials have been conducted to investigate isotypes of PD-L1 therapies, such as Pembrolizumab, which was approved by the FDA in May 2017 to treat MSI-H/dMMR advanced CRC patients that progressed on conventional chemotherapy. The study was based on the outcomes from the clinical output of the phase II trial, KEYNOTE-028 (FDA Grants Pembrolizumab). Pembrolizumab was also approved as the first-line treatment for patients with metastatic CRC (mCRC) and MSI-H/dMMR, as the outcomes from the KEYNOTE-177 phase III study [22]. Furthermore, CheckMate-142, a phase II open-label trial, investigated Nivolumab, a human IgG4 mAb, a PD-1 inhibitor, that led to the FDA approval of Nivolumab to treat MSI-H/dMMR mCRC with progressive disease after chemotherapy (FDA Grants Nivolumab). In total, there are currently ten mAbs approved targeting PD-1, namely Nivolumab, Avelumab, Cemiplimab, Pembrolizumab, Toripalimab, Sintilimab, Camrelizumab, Tislelizumab, Zimberelimab, and Prolgolimab, and three mAbs targeting PD-L1: Atezolizumab, Durvalumab, and Avelumab [23].

Moreover, in the neoadjuvant setting, Pembrolizumab was administered to Lynch syndrome patients, who qualified for surgical resection afterward [24]. Zhang et al. investigated two patients with locally advanced CRC and proved that Nivolumab in the neoadjuvant setting can generate full positive responses, either as a single therapy option or followed by surgery [25]. Neoadjuvant Immune Checkpoint Inhibition and Novel Immuno-oncology (IO) Combinations in Early-stage Colon Cancer (NICHE) clinical trials demonstrated that Nivolumab induced a significant response as a neoadjuvant treatment for early-stage CRC [26]. All of the above emphasize the potential role of anti-PD-1 in the adjuvant and neoadjuvant CRC treatment.

Table 1 summarizes the clinical trials in which immune checkpoint monoclonal antibodies against PD-1/PD-L1 were combined with different lines of conventional therapies in MSI-H/dMMR CRC patients. The completed clinical trial, as seen in Table 1, concludes that Pembrolizumab has a good safety profile and is effective in patients with MSI-H/dMMR CRC and other MMR-deficient cancers due to the great proportion of mutant neoantigens that are sensitive to immune checkpoint blockades, irrespective of the origin of cancerous cells [27,28]. Intriguingly, the efficiency of the Varlilumab and Nivolumab combination was not higher than monotherapy with Nivolumab [29]. Regarding the non-completed clinical trial shown in Table 1, the preliminary finding exhibited substantial advancements in the treatment of MSI-H/dMMR CRC. It accentuated the importance of mismatch repair status as a reliable predictor for the clinical benefit of immune checkpoint blockade therapies in patients [30], while Table 2 summarizes the clinical trials in which immune checkpoints against PD-1/PD-L1 were combined with different lines of conventional therapies in MSS/pMMR CRC patients. The outcomes are not yet optimal, and the results highlighted the difficulty of extending immunotherapy benefits to microsatellite-stable CRC patients with immunologically cold tumors, as their tumor microenvironments have a lower number of baseline infiltrating immune cells than microsatellite-instable CRC patients [31,32].

**Table 1 cells-14-00745-t001:** Immune checkpoint inhibitors (Anti-PD-1/PD-L1) in clinical trials for MSI-H/dMMR CRC.

MSI-H/dMMR CRC Phenotype	Immune Target	Treatment Combination	Study Phase	Clinical Trial Identifier	Status	Outcome
mCRC MSI-H/dMMR	PD-1	Pembrolizumab	II	NCT02460198	Completed	ORR = 32.8% [28,33,34]
MSI-H/dMMR tumors	PD-1	Pembrolizumab	II	NCT01876511	Completed	(ORR) = 50% [27,30]
MSI-H/dMMR CRC	PD-1	Pembrolizumab + Epacadosat	I/II	NCT02178722	Completed	ORR(I) = 57% ORR(II) = 80%
Advanced (CRC)	PD-1	Varlilumab + Nivolumab	I II	NCT02335918	Completed	ORR = 5% [29]
MSI-H/dMMR and MSS/pMMR mCRC and MSI-H/dMMR endometrial carcinoma	PD-1	Pembrolizumab + Ataluren	I-II	NCT04014530	Recruiting	ORR = 71% [30]
MSI-H/dMMR CRC	PD-1	Pembrolizumab + COX inhibitor (aspirin)	II	NCT03638297	Recruiting	N.A.
MSI-H/dMMRmetastatic solid tumors	PD-1	Pembrolizumab + RT (metastatic site) vs. Pembrolizumab	II	NCT04001101	Recruiting	N.A.
MSI-H/dMMR mCRC	PD-1	Nivolumab + Ipilimumab	II	NCT04730544	Recruiting	N.A.
MSI-H/dMMR mCRC	PD-L1	Avelumab	II	NCT03186326	Recruiting	N.A.
Advanced or metastatic solid tumors, including MSI-H/dMMR CRC	PD-L1	Avelumab + Regorafenib	I-II	NCT03475953	Recruiting	N.A.
MSI-H/dMMR or POLE mutated mCRC	PD-L1	Durvalumab	II	NCT03435107	Active, not recruiting	N.A.
Advanced pancreatic cancer NSCLC dMMR CRC	PD-L1	Danvatirsen + Durvalumab	II	NCT02983578	Active, not recruiting	N.A.
Advanced MSI-H/dMMR CRC	PD-L1	Durvalumab	II	NCT02227667	Completed	N.A.
Metastatic/ advanced CRC and PaC	PD-L1	Durvalumab + Pexidartinib	I	NCT02777710	Completed	N.A.
Advanced chemotherapy-resistant MSI/dMMR CRC	PD-L1	Atezolizumab + Bevacizumab	II	NCT02982694	Recruiting	N.A.
MSI-H/dMMR mCRC	PD-L1	Atezolizumab vs. Atezolizumab + Bevacizumab + FOLFOX	III	NCT02997228	Recruiting	N.A.
MSI-H/dMMR mCRC	PD-1 + CTLA-4	Nivolumab vs. Nivolumab + Ipilimumab Nivolumab + Ipilimumab vs. chemotherapy	III	NCT04008030	Recruiting	N.A.
MSI-H/dMMR CRC, MMS CRC, pancreatic cancer	PD-1 + CTLA-4	Nivolumab + Ipilimumab + RT	II	NCT03104439	Recruiting	N.A.
Recurrent or metastatic MSI-H and non-MSI-H CRC	PD-1 + CTLA-4	Nivolumab Nivolumab + Ipilimumab Nivolumab + Ipiliumab + Cobimetinib Nivolumab + BMS-986016 Nivolumab + Daratumumab	II	NCT02060188	Active, not recruiting	N.A.

dMMR: mismatch repair deficiency, MSI-H: microsatellite instability-high, MSS: microsatellite stable, mCRC: metastatic colorectal cancer, FOLFOX: leucovorin calcium (folinic acid), fluorouracil, and oxaliplatin, NSCLC: non-small cell lung cancer, PaC: pancreatic cancer, ORR: objective response rate, N.A.: not available, R: refractory colorectal cancer.

**Table 2 cells-14-00745-t002:** Immune checkpoint inhibitors (Anti-PD-1/PD-L1) clinical trials for MSS/pMMR CRC.

MMS/pMMR CRC Phenotype	Immune Target/Generic Name	Treatment Combination	Study Phase	Clinical Trial Identifier	Status	Outcome
advanced MSS/pMMR mCRC	PD-L1 + PD-1	Regorafenib + Nivolumab	II	NCT04126733	Completed	ORR = 33% [35]
mCRC and pancreatic cancer	PD-1	Olaptesed pegol + Pembrolizumab	I/II	NCT03168139	Completed	N.A.
Advanced MSS/pMMR CRC	PD-1	Pembrolizumab + cyclophosphamide + Colon cancer vaccine	II	NCT02981524	Completed	ORR = 1.6% Did not meet its primary objective in MSS/pMMR CRC [36]
Refractory MSS/ pMMR mCRC	PD-1	Pembrolizumab + Maraviroc	I	NCT03274804	Completed	Therapy combination is feasible with a beneficial toxicity pattern [37]
MSS/pMMR CRC	PD-1	Nivolumab + Tipiracil hydrochloride	II	NCT02860546	Completed	Therapy combination is feasibly tolerable. No clinical benefit to MSS, mCRC failed [38]
MSS/pMMR CRC	PD-L1	Avelumab + Tomivosertib vs. Tomivosertib	II	NCT03258398	Completed	N.A.
Chemorefractory MSS/pMMR mCRC	PD-1	Pembrolizumab + Azacitidine	II	NCT02260440	Completed	ORR = 3% The therapy combination is safe and tolerable with modest clinical activity [39]
MSS/pMMR mCRC mPaC	PD-1	Olaptesed pegol vs. Olaptesed pegol + Pembrolizumab	I/II	NCT03168139	Completed	N.A.
Advanced solid tumors (including MSS/pMMR CRC)	PD-L1	Azacitidine + Durvalumab	II	NCT02811497	Completed	Did not show strong effects in immunologically cold solid tumors [32]
MSS/pMMR mCRC (Liver)	PD-L1	Durvalumab + Tremelimumab following radioembolization (RE) with SIR-spheres	I	NCT03005002	Completed	N.A.
Non MSI-H mCRC	PD-L1	Cobimetinib + Bevacizumab + Atezolizumab	Ib	NCT02876224	Completed	N.A.
mCRC	PD-L1	Atezolizumab (A) vs. Atezolizumab (A) + Cobimetinib (C) vs. Regorafenib	III	NCT02788279	Completed	Therapy combination (B and C) did not improve overall survival. Safety of the therapy combination is consistent with the individual drugs

dMMR: mismatch repair deficiency, MSI-H: microsatellite instability-high, MSS: microsatellite stable, mCRC: metastatic colorectal cancer, FOLFOX: leucovorin calcium (folinic acid), fluorouracil, and oxaliplatin, NSCLC: non-small cell lung cancer, ORR: objective response rate, N.A.: not available, R: refractory colorectal cancer.

To sum up, ongoing clinical trials are trying to improve the use of ICIs as first-line adjuvant therapies for MSI-H/dMMR CRC by understanding resistance pathways and potential methods to overcome them, as well as investigating strategies to modify the tumor microenvironment to enhance tumor recognition by the immune system, to include more candidates of MSS/pMMR CRC patients. The next section focuses on the study of the proposed mechanisms of resistance to anti-PD-1/PD-L1 treatment.

### 2.2. Current Limitations of Anti-PD-1/PD-L1 Therapy in Colorectal Cancer

While immunotherapy targeting the PD-1/PD-L1 axis shows promising outcomes in the treatment of different types of cancer, including melanoma, non-small cell lung cancer, and renal cell carcinoma, its effectiveness in patients with other solid tumors is limited [40]. For instance, the administration of PD-1/PD-L1 to MSI-H/dMMR CRC patients is considered relatively recent and has shown a satisfactory outcome, but resistance to these treatments has already been documented [41]. The success of the PD-1/PD-L1 blockade in treating tumors hinges on the presence of antigen-specific T-cell reactivity within the tumor microenvironment. This reactivity is reliant on the presentation of potential tumor rejection antigens by dendritic cells to prime CD8+ T-cells [42]. Subsequently, this priming process triggers a cascade of events and leads to the initiation of antitumor activity [42]. This process is critical in promoting the body’s immune response against the tumor [43].

Kloor et al. suggested one of the mechanisms that cancer cells use to escape immune surveillance, namely, to alter the human leukocyte antigen (HLA) complex expression. This alteration leads to a decrease in the ability to process and present antigens on MHC molecules, making it harder for the immune system to recognize and attack the cancer cells [44]. Grasso et al. conducted a cohort investigation on the potential immune evasion mechanism of 179 MSI-H/dMMR CRC patients [45]. The study identified that most MSI-H/dMMR tumors harbor at least one mutation in the genes related to the immune response, such as those involved in B-cell development, T-cell response, and natural killer (NK) cell function, that could impair antigen presentation. However, these initial mutations were not sufficient to confer resistance to ICIs. Therefore, they suggest that “immune editing is preceding the treatment and tumors are on a resistance continuum”. Furthermore, the persistence of antitumor immune response depends on tumor-specific antigen expression. When it comes to CRC tumors, due to frameshift mutations resulting from MMR deficiency (MSI-H/dMMR tumors), they generate about 50 times more neoantigens than MSS/pMMR tumors [46]. This highlights important questions regarding the quality of mutations. This means that point mutations, which result in minor changes to the amino acid sequence in a protein’s structure, are less likely to trigger a strong immune response when compared to mutations that impact the antigenic properties of proteins [47]. However, loss of expression of the MMR gene may not always indicate microsatellite instability status. For instance, patients may have an MSI-L/dMMR tumor that mimics the MSS/pMMR phenotype. As a result, these patients may not respond adequately to PD-1 therapy [48]. Identifying biomarkers of response has emerged as a hot research topic.

Several reports indicated that the poor outcome of PD-1 therapy in MSI-L/pMMR tumors might be because of the highly infiltrated myeloid-derived suppressor cells (MDSCs) and regulatory T-cells (Tregs) in this subtype of tumors when compared to MSI-H/dMMR ones [49]. Myeloid-derived suppressor cells are key players in tumor immune evasion, producing enzymes like nitric oxide synthase and arginase-1, which deplete L-arginine, an amino acid essential for T-cell function. This L-arginine depletion impairs T-cell activation and metabolism, weakening the immune response. MDSCs also release immunosuppressive cytokines such as interleukin-10 (IL-10) and transforming growth factor-beta (TGF-β), further inhibiting T-cell proliferation and creating an immunosuppressive environment that protects tumors from immune surveillance. Together, these effects significantly hinder the body’s ability to launch an effective anti-tumor response [50].

Intriguingly, specific mutations have been shown to convey resistance to immunotherapy. For example, mutations in the β-2 microglobulin (*B2M*) gene, an important part of the HLA-class I complex, are another resistance mechanism to immunotherapy, as they result in complete loss of HLA class I molecules on the cell surface [51]. *B2M* somatic mutations occur very often in the coding microsatellites as a result of microsatellite instability and were found in about 30% of the MSI-H/dMMR CRCs and less than 2% in MSS/pMMR tumors [27,52].

It is well known that the interferon–gamma (IFN-γ) signaling pathway plays a crucial role in regulating the expression of PD-L1 through the action of Janus kinases 1 and 2 (JAK1 and JAK2) [53,54]. Mutations in JAK1 and JAK2 significantly downregulate the PD-L1 gene expression in MSI-H/dMMR CRCs, melanoma, and endometrial tumors, leading to resistance to PD-1 therapy [55,56]. Mutations in signal transducer and activator of transcription proteins (STATs 1/2), in the downstream JAK-signaling pathway, are another potent mediator of IFN-γ. Therefore, these mutations could generate impaired IFN-γ signaling and eventually immune escape and immunotherapy resistance [57]. All the discussed immunotherapy resistance mechanisms are illustrated in Figure 1.

In conclusion, the interest in PD-1/PD-L1 blockades for treating CRC is growing, particularly as it is considered the primary area of immunotherapy in MSI tumors. However, it is important to note that 85% of colorectal tumors are MSS, and so far, immunotherapy has yet to yield substantial clinical outcomes in these cases. Additionally, the resistance to therapy necessitates the development of new treatment strategies to improve colorectal cancer immunotherapy. In the following section, we will highlight the role of autophagy in CRC and suggest its modulation as a promising target for enhancing the effectiveness of immunotherapy for CRC treatment.

## 3. Autophagy as a Modulator of Immune Response in Cancer

Autophagy is a major self-regulatory mechanism that is involved in preserving cellular homeostasis by turning over long-lived proteins and ingesting abnormal cell organelles, as explained by Koustas et al. and Davenport et al. [58,59]. Functionally, autophagy is classified into three major forms: Chaperone-Mediated Autophagy (CMA), micro-autophagy, and macro-autophagy [60]. All forms of autophagy are lysosome-dependent, but they differ in the regulatory mechanisms and the conditions under which it is triggered [61,62]. For instance, the selective and nonselective landscape of engulfment of cytoplasmic contents of micro- and macro-autophagy has been broadly characterized [63].

The key utility of non-selective or basal autophagy is to support cell survival during inadvertent microenvironment conditions such as nutrient starvation, oxygen deprivation (hypoxia), or metabolic stressors, namely cellular reactive oxygen species (ROS) [64,65]. On the other hand, selective autophagy sustains cellular homeostasis by selectively identifying, targeting, and eliminating specific organelles such as mitochondria (mitophagy), peroxisomes (peroxyphagy), ribosomes (ribophagy), endoplasmic reticulum (reticulophagy), protein aggregates (aggrephagy), or infectious pathogens (xenophagy) [63,66]. The end bioproducts of autophagy, such as amino acids, nucleotides, and fatty acids, are recycled and used in the synthesis of new cellular components within the cell itself [67,68]. Interestingly, studies have shown that the contents of autophagosomes can also be secreted outside the cell in a process called secretory autophagy for signaling purposes or to eliminate intracellular debris [69,70]. This process can modify the microenvironment, as will be explained in the following sections [71,72].

### 3.1. The Role of Autophagy in the Tumor Microenvironment

The CRC, as a solid tumor, is highly heterogeneous at both cellular (mesenchymal cells, endothelial cells, fibroblasts, and immune cells) and acellular (extracellular matrix proteins, such as collagen, elastin, fibronectin, laminin, and secretory proteins, including cytokines, chemokines, proteases, growth factors, and metabolites) levels, which contribute to the formation of the tumor microenvironment (TME) [73]. The CRC TME is characterized by low pH, hypoxia, and high metabolites [74]. Energy acquisition in such environments is hard, considering that tumor cells need to meet bioenergetic requirements for cancer biosynthetic growth and proliferation [75]. Here, cancer cells will rewire metabolism and become more dependent on the glycolytic pathway, resulting in higher autophagy.

Glycolysis is a ten-step, consecutive metabolic process that occurs in the cytoplasm to provide ATP by converting glucose to pyruvate. In an aerobic setting, pyruvate is subjected to oxidative phosphorylation to produce 36/38 ATP [76]. Nevertheless, tumor cells survive in anaerobic conditions, so they process pyruvate to lactate and produce only two ATPs, a phenomenon described in the 1920s by Otto Warburg [77,78]. Consequently, under starvation, autophagy is directly upregulated through phosphorylation of unc-51-like kinase 1 (Ulk1) from upstream sensors mTOR and AMPK [79], and either increases mitochondrial oxidative metabolism or complements glycolysis by providing raw material after digesting complex proteins and lipids in tumor cells experiencing metabolic stress, depending on their type [67,80]. On one hand, stromal cells exhibit the Warburg Effect and parasitically supply glucose to adjacent epithelial cancer cells, which utilize oxidative phosphorylation for energy production. On the other hand, epithelial cancer cells produce ROS and transmit them to the adjacent stromal cells, creating oxidative stress and upregulating autophagy. Cancer-associated fibroblasts (CAFs) upregulate secretory autophagy to release high-energy recycled nutrients such as ketone bodies, L-lactate, and glutamine. The latter are utilized for oxidative mitochondrial metabolism in epithelial cancer cells [81]. Figure 2 describes the dynamic interaction between glycolysis enzymes and autophagy regulation.

Moreover, immunogenic cell death of cancer cells can only occur in cells with a functioning autophagy system that can keep a supply of ATP in the lysosomes. Lysosomal fusion with the plasma membrane allows ATP and antigens to be released on the tumor cell surface, leading to the recruitment of dendritic cells, macrophages, and monocytes, hence boosting their anticancer activities [69,72].

To sum up, autophagy modulates the TME’s generation during carcinogenesis, and this unique microenvironment alters the autophagy signaling pathways in innate immune cells, stromal cells, and cancerous cells [82]. Consequently, autophagy may be utilized to enhance the innate immune system as well as the efficacy of immunotherapy to combat cancer.

### 3.2. Autophagy and Antigen Presentation

Autophagy proteins interact with different pathways of dendritic cell antigen presentation to T-cells [83,84]. The study conducted by Paludan et al. was the first to provide evidence that autophagy can deliver endogenously synthesized antigens for presentation on MHC class II molecules to CD4+ T-cells [85] (Figure 3). This study investigated whether autophagy promotes endogenous MHC class II processing of the Epstein–Barr virus’s nuclear antigen 1 (EBNA1). EBNA1 is the predominant EBV-latent antigen for CD4+ T-cells and can be identified by CD4+ T-cells following endogenous MHC class II processing in EBV-positive lymphoma cells. Treatment with the autophagy inhibitor 3-methyladenine (3-MA) for 2 to 4 days reduced EBNA1-specific CD4+ T-cell recognition of EBV-transformed B-cells and EBNA1-transfected Hodgkin’s lymphoma cells by 30% to 70% in interferon–gamma enzyme-linked immunospot (ELISPOT) assays [85].

Moreover, Münz et al. and English et al. suggested that autophagy enhances the intracellular antigen presentation to CD8+ T-cells via MHC class I [86,87]. Macro-autophagy appears to be required in the antigen donor cell for efficient cross-presentation on MHC class I molecules, possibly by assisting exosome generation after autophagic delivery of antigens to multivesicular bodies (MVBs) [87]. MHC-I expression is elevated in *ATG5*- and *ATG7-deficient* dendritic cells owing to decreased endocytosis and degradation, meaning that the absence of the macro-autophagy attenuated internalization of MHC class I molecules and increased their stability on the cell surface, whereas they kept a normal migration and innate responses [88,89]. Moreover, autophagy affects the degradation of major histocompatibility complex I and immune cell infiltration [90].

Yamamoto et al. suggested that autophagy may be involved in tumor progression via an immune mechanism. In this regard, they observed that NBR1, a crucial autophagosome content receptor in solid tumors, plays a central role in the expression of MHC-I on the surface of the tumor cells, as well as within autophagosomes and lysosomes. Given the significant connection between MHC-I and antigen presentation, vital for the anti-tumor activity of immune cells, intervening with NBR1 emerges as a potential avenue to modulate autophagy. By doing so, it influences the expression of MHC-I, suggesting that autophagy could promote tumor progression through an immune mechanism by intervening with NBR1 [90]. Vacuolar protein sorting 34 (VPS34) is another autophagy protein. In mice, when it was obliterated, the DCs’ presentation of MHC-I and MHC-II antigens was increased [91]. This might be due to VPS 34’s role in autophagy induction by forming a complex with BECN1/Beclin to initiate autophagy and participate in autophagosome formation [92]. Interestingly, in myeloid-derived suppressor cells, the membrane-associated RING-CH-type finger (MARCH) E3 ubiquitin ligase induces autophagic degradation of MHC-II [89].

### 3.3. Tumor Cell Autophagy

Within the restricted tumor microenvironment, cancer cells utilize autophagy to sustain the unavailability of essential survival elements [93]. Autophagy is initiated and maintained by various genes and proteins, including *Beclin-1*, *LC3*, *ATG5*, and *ATG7*. These autophagy-related genes vary in expression in cancer cells, including colorectal cancer, compared to normal cells [94]. For instance, increased *Beclin-1* expression levels are negatively associated with metastasis and related to a favorable outcome. Moreover, decreased expression levels of *Beclin-1* are related to increased survival in advanced cases treated with cetuximab, and it is associated with a good response after chemoradiation in patients with rectal cancer [95,96]. The LC3-II levels have prognostic significance in CRC, as their increased expression, particularly in advanced stages, is related to aggressiveness, whereas decreased expression levels are related to good outcomes and treatment responses [97,98].

Frameshift mutation of several *ATG* genes, such as *ATG2B*, *ATG5*, *ATG9B*, and *ATG12*, is observed in human cancer, which may contribute to tumorigenesis [82,99]. Burada et al. reported a higher expression of the aforementioned genes in more aggressive CRC phenotypes [98]. These findings have led to more research to support the role of autophagy in CRC development. Research findings suggest that because autophagy is responsible for maintaining the energy requirement during critical cell activities such as proliferation, epithelial-mesenchymal transition, migration, and angiogenesis, it appears to be directly linked to general tumor growth [100]. Autophagy is upregulated in various regions of the already established tumors [101]. Several studies discuss the role of autophagy in therapy resistance [102]. And it has also been determined that autophagy negatively impacts cancer patients’ response to chemotherapy, as shown in Table 3. High levels of autophagy are linked to dismal survival rates and poor response to therapeutic drugs [103]. Despite all the previous evidence, the role of autophagy in CRC Table 4 and cancer, in general, is more complex and context-dependent on the tumor type, genetic and epigenetic status, and tumor stage/site-specific manner [104].

In other words, the role of autophagy in CRC is paradoxical and can serve as both pro- and anti-tumor [105,106]. In the early stages of tumorigenesis, autophagy has an antitumor effect orchestrated by limiting chromosomal instability, restraining necrosis and inflammation, and promoting senescence [104,107]. However, in more advanced cancers, autophagy is directly linked to the progression of the tumor [108,109].

Several studies discuss the role of autophagy in therapy resistance [102]. Table 3 explores the effect of autophagy inhibitors on drug-resistant cancer cells [58,105,110,111,112,113,114].

**Table 3 cells-14-00745-t003:** Autophagy inhibitors effect on drug-resistant cancer cells.

Cancer	Drug Resistance Mediated by Autophagy Induction	Autophagy Inhibitor	Mechanism of Targeting Drug-Resistant Cancer Cells	Reference
Colorectal Cancer	PFKFB3 inhibitor, 3PO	3-methyladenine/Chloroquine	Inhibition of autophagy induced due to PFKFB3 inhibition	[114]
Colorectal Cancer	Cabozantinib XL184	SBI0226365/Chloroquine	Inhibition of autophagy-dependent metabolism	[112]
Colon Cancer	Inhibition of ANKRD37	Chloroquine	Inhibition of autophagy is induced due to ANKRD37 translocation to the nucleus	[113]
Colon Cancer Cells	CoCl2	3-methyladenine	Inhibits hypoxia-induced autophagy	[105]
Colon Cancer	NA	3-methyladenine	Inhibits the supply of free fatty acid (FFA) from adipocytes	[111]
Colon adenocarcinoma	Oxaliplatin	SP600125	JNK inhibition prevents hypoxia-induced autophagy	[110]

PFKFB3: 6-phosphofructo-2-kinase/fructose-2,6-biphosphatase3, JNK: c-Jun N-terminal kinase, LDHA: lactate dehydrogenase A, HIF1α: hypoxia-inducible factor 1 subunit alpha, NF-κB: nuclear factor kappa-light-chain-enhancer of activated B-cells, CoCl2: Cobalt (II) chloride.

### 3.4. Autophagy in Immune Cells and Others

The activation of effector T-cells leads to the initiation of macro-autophagy, due to dramatic demand for glucose metabolism and other bioenergetic requirements such as fatty acids, and glutamine [115]. As a result of the activation of both the T-cell receptor as well as CD28, there is an increased processing of LC3 [116]. An increase in LC3-containing vesicles and LC3 flux are considered as signs of active autophagosome formation and clearance through fusion with lysosomes to degrade their cargo [115]. To study the role of autophagy in adaptive immunity, researchers have knocked out core autophagy genes (*Atg3*, *Atg5*, *Atg7*, *Atg16L1*, *Becn1*, or *Pik3c3/Vps34*) in mice [117,118]. Collectively, these studies elucidated that ablation of the core autophagy proteins resulted in a significant reduction in thymocytes and peripheral T-cells (lymphocyte survival) [119,120]. This happens in autophagy-defective T-cells, especially during the transition from thymocytes to mature circulating T-cells, increasing ROS production in the endoplasmic reticulum, and accumulating damaged mitochondria, due to defective clearance of mitochondria via mitophagy and reticulophagy. On the other hand, autophagy-defective peripheral T-cells have a higher apoptotic rate due to the upregulation of proapoptotic molecules, such as pro-CASP3 (caspase 3), CASP8, and CASP9 [121]. Interestingly, Nedjic et al. suggested that in thymic epithelial cells, ablation of *ATG5* leads to the altered elimination of autoreactive T-cells, which restricts T-cell specificities and autoimmunity known as thymic negative selection [122]. Interleukin-1β (IL-1β) and interleukin-18 (IL-18) are multifunctional pro-inflammatory cytokine that affects both the innate and adaptive immune systems, are secreted through secretory autophagy, and are heavily influenced by autophagy induction circumstances. Tumor-released autophagosomes (TRAPs) in the microenvironment polarize macrophages to an M2-like pro-tumor phenotype, yielding immunosuppressive functions [123]. The following section will address this by examining the different in vivo models for studying immunotherapy/PD-1/PD-L1 therapy and monitoring autophagy for better therapeutic outcomes.

## 4. Preclinical Models for Testing Anti-PD-1/PD-L1 Therapies

Developing realistic preclinical models that mirror the tumor microenvironment is important in cancer therapeutics [124]. Cell line cultures have guided the generation of preclinical data, but the genetic differences between these and their corresponding primary tumors have resulted in suboptimum clinical translation. Immune checkpoint inhibitors have revolutionized cancer treatment because of their safety profile and their promising outcome over a long duration [125,126]. Unfortunately, their clinical response is cancer-type and stage-dependent and subject to individual variation [127]. In this section, we review the available preclinical models to evaluate the efficacy of immunotherapies in a signed context. Moreover, we will briefly go through monitoring autophagy in vivo to accurately document the process to a satisfactory outcome. The ideal model should be scalable, reliable in the treatment outcomes, flexible to be used with different models, and reproducible over time [128]. Each of these ICIs has a unique binding site on its target molecule and different modifications to the fragment crystallizable (Fc) region of the antibody [125,129,130]. Structural modeling and conformational changes of the receptor control the ICI’s binding affinity considerably, and hence, the variation in their subsequent signaling pathways resulted in different outcomes [131,132]. In sum, a key hurdle toward the advancement of these therapies is the availability of immunocompetent preclinical models that recapitulate human disease to understand their interaction and hence predict their potential therapeutic outcome. On the other hand, monitoring the autophagy flux process in vivo or organs is one of the least developed areas until this moment, and ideal techniques that work in cell culture may not exist or cannot be extrapolated to an in vivo setting. Additionally, the level of basal autophagy and reliable negative control, the time course of autophagic induction, and the bioavailability of autophagy-stimulating and -inhibiting drugs are likely organ- and tissue-specific [133]. Moreover, basal autophagy or sensitivity to autophagic induction or inhibition may vary with model age, sex, or strain genetic background. Therefore, methods may need to be optimized and tuned carefully for the tissue or organ, or tumor of interest [133].

### 4.1. Clinical Outcome for Anti-PD-1/PD-L1 Using Different Models

In this part of the review, we want to highlight the central concept of designing a suitable model for testing anti-PD-1/PD-L1 therapies: the idea of activating T-cells and how to create a response related to PD-1/PD-L1. Moreover, the dynamics of using syngeneic models or humanized mouse models with activation of the immune system should be seen to elicit their anti-tumor therapeutic efficacy. Bareham et al. developed ‘HuPD-H1’ mice that express humanized PD-1/PD-L1 to allow the use of anti-PD-1/ PD-L1 *in vivo.* They genetically engineered mice expressing PD-1 and PD-L1 that could be coupled with the approved humanized monoclonal antibodies. These mice were immunocompetent C57BL/6 strain normal, with no marked differences compared to wild-type C57BL/6 [134]. Their results proved that HuPD-H1 mice can be a promising tool to shorten the gap between the preclinical and clinical studies to improve and optimize the anti-PD-1/PD-L1 therapeutic use [125]. In the future, these mice will serve as reasonable models to evaluate different combinatory therapeutic modalities with conventional anticancer drugs, small molecules, and radiation therapy. In addition, they can elucidate immune cell biomarkers linked to the response of PD-1/PD-L1 inhibitors [125].

In the same context, Shang et al. developed a cell line-derived xenograft humanized mouse model to study the PD-1/PD-L1 inhibitors in colorectal cancer. They constructed humanized NCG mice by transplanting human CD34+ stem cells into non-obese diabetic mice and monitoring the development of human hematopoietic and immune systems (Human-NCG). Their mouse models are designed to evaluate the outcome of immunotherapy mediated by the BMS202 (a PD-1/PD-L1 inhibitor), which shows hopeful antitumor efficacy in colorectal cell line xenograft mouse models. It exerts anticancer efficacy by enhancing the tumor microenvironment and improving the infiltration of human CD8+ T-cells and the release of human IFNγ in tumor tissue. Hence, their mouse models are suitable preclinical models for studying immune system surveillance and immune evasion by cancerous cells [135].

### 4.2. Study of Autophagy in Animal Models

For the in vivo investigation of autophagy, two types of gene-modified mouse models have been used to monitor autophagy and evaluate the genetic autophagy inhibition outcome [136,137]. In this regard, transgenic mice systemically expressing autophagy markers like LC3 are used [136,138]. Utilizing GFP-LC3 mice, observation and quantification of autophagy in vivo became simple and feasible. Another approach that has greatly contributed to comprehending the role of autophagy in vivo is mice deficient in specific *Atg* genes [136]. A study by Levy et al. illustrated the role of autophagy in colorectal cancer. As is known, CRC is frequently characterized by a high mutational burden. Subsequently, autophagy was inhibited in an APC^+/−^-driven mouse model of CRC. Autophagy deficiency prevented tumor progression and reduced the number of precancerous polyps. This anti-cancer effect was mainly mediated by the cytotoxic CD8+, IFN, and cytotoxic T-cells, and partially due to metabolic defects, p53-mediated cell cycle arrests, and cell death [136,137,138,139,140,141].

This part emphasizes how the proper selection of preclinical models is a fine-tuning process, whether for testing the immunotherapies or autophagy flux itself in animal models. The selection is based on many aspects related to genetic profiles, disease heterogenicity, and the anticipated mode of action of the used immunotherapies or autophagy regulators drugs to induce or inhibit autophagy at its different stages from initiation to autophagolysosomal conjugation. Studies highlighted that the use of anti-PD1/PD-L1 therapies and autophagy modulators in combinatory models is still limited. Extra precautions are needed to achieve the desired outcome. All the reported combination models utilize syngeneic models, and autophagy was monitored afterward by measuring the autophagy-related proteins or pathways as LC3B, p62, etc. [142]. These approaches still confer limitations to accurately interpret the findings to leverage such combos into the clinical setting and, hence, study their potential therapeutic value [133,142].

## 5. Therapeutic Perspectives of Targeting Autophagy to Enhance the Response of PD-1/PD-L1 Therapy

PD-1/PD-L1 pathways control the induction and maintenance of immune tolerance within the tumor microenvironment, as seen in Figure 4. They are responsible for T-cell activation and proliferation [143]. On one hand, studies have demonstrated that autophagy can be a prospective strategy that boosts the immune response and antitumor effect of anti-PD-1/PD-L1 therapies, as it enhances antigen presentation and sensitivity to cytotoxic T-lymphocytes (CTLs), especially CD8+ T-cells [79,144]. As mentioned before and elucidated in Figure 3, autophagy is involved in antigen processing for MHCI and MHCII presentations. Li et al. and Hahn and Akporiaye et al. investigated the role of alpha-tocopheryloxyacetic acid (α-TEA), a semisynthetic vitamin E derivative, as an adjuvant strategy to improve anti-PD-1/PD-L1 therapy. They found that α-TEA can stimulate autophagy to strengthen MHCI cross-presentation of tumor antigens, reinforcing CD8+ T-cells’ anti-tumor immune response [145,146].

Autophagy can also reduce the expression of PD-L1 in vitro and in vivo, thus suppressing cancer cells’ ability to evade immune surveillance [147]. On the other hand, other studies suggested that autophagy inhibition in tumor cells can enhance anti-PD-1/PD-L1 therapeutic efficacy. Cancer cells can prevent PD-L1 autophagic degradation by transcriptional change of oncogenic pathways such as Myelocytomatosis oncogene (MYC), anaplastic lymphoma kinase (ALK), HIF1α, NF-κB, Mitogen-activated protein kinases (MAPK), Phosphatase and tensin homolog (PTEN)/PI3K, and epidermal growth factor receptor (EGFR) [144,148]. In the colon cancer model, DHHC3’s palmitoylation of PD-L1 was reported to limit the autophagic degradation of PD-L1 due to endosomal sorting, leading to immune response suppression and tumor progression [144]. Similarly, in a breast xenograft tumor model, it was found that PD-L1 glycosylation mediated by the EGFR/B3GNT3 pathway suppresses the autophagic degradation of PD-L1, and in turn, promotes tumor immune escape [148].

Zhang et al. investigated an in vivo model using an *ATG7* knockdown MSI-H/dMMR CRC cell line. CD8+ T effector cells were found to be infiltrating and inhibiting tumor growth. Also, increased surface MHC-I levels lead to better antigen presentation and anti-tumor T-cell response through the ROS/NF-κB pathway. Therefore, autophagy inhibition can improve the therapeutic benefit of anti-PD-1 drugs in MSI-H/dMMR CRC [149]. In addition, hypoxia-induced autophagy inhibits NK-mediated killing and degrades NK-derived Granzyme B (GrzB), reducing immunotherapy effectiveness by decreasing CTLs-mediated tumor cell lysis and pSTAT3 phosphorylation [150,151].

In conclusion, autophagy can either promote cell survival (cytoprotective) in stressful conditions or lead to cell death (cytotoxic). The impact of autophagy inhibition may vary depending on whether autophagy is functioning protectively or destructively in a particular cancer [152]. Highlighting the responses to this innovative therapy in Table 4, hydroxychloroquine (HCQ), the only FDA-approved autophagy inhibitor, has an immunomodulatory effect and has been used for the treatment of rheumatic arthritis and other autoimmune diseases. Along with its antimalarial and anticancer effects, it can interfere with multiple pathways. The inhibition of autophagy by HCQ has been documented at various dosage levels between 10 and 80 mg/kg [153]. When using HCQ as an autophagy inhibitor, the researchers are divided into two groups; one suggests that its autophagy inhibition compromises the success of immunotherapy by introducing significant immunosuppressive effects. For example, Wabitsch et al. thoroughly investigated the in vitro and in vivo impacts of (HCQ) treatment on the PD-1 therapeutic efficacy using anti-PD-1-sensitive tumor cell lines (MC38, CT26, and RIL-175). Firstly, in vitro experiments revealed that HCQ exerts a direct inhibitory effect on tumor cell growth across all three cell lines. Secondly, in in vivo models, HCQ treatment decreased the activation of T-cells, as it significantly impaired the production of immune signaling molecules TNFα and IFNγ in both antigen-specific and nonspecific T-cells. Also, they described a notable reduction in the infiltration of tumor-targeting, antigen-specific CD8+ T-cells within the tumor microenvironment. This study demonstrated the broader immunosuppressive effects of HCQ treatment by counterbalancing the beneficial increase in MHC-I expression facilitated by tumor cells and undermining the direct cytotoxic responses intended by the anti-PD-1 treatment in both in vitro and in vivo models [154]. In the same context, Krueger et al. assessed HCQ’s potential influence on tumor dynamics, finding that the administration of HCQ in vivo inhibited PD-L1 expression on tumor cells, leading to a pronounced reduction in the effectiveness of anti-PD-1 therapy [155].

The second group of researchers suggested that autophagy inhibition augments the antitumor efficacy of anti-PD-1 treatment, as shown in Table 5. For instance, Sharma et al. examined palmitoyl-protein thioesterase 1 (PPT1), focused on in vivo models; the findings exhibited enhanced anti–PD–1 therapeutic response in melanoma prompted through three mechanisms: (1) it facilitates a switch from M2 to M1 macrophage polarization, (2) diminishes the presence of myeloid-derived suppressor cells (MDSCs) in the tumor microenvironment, (3) triggers the release of IFN-β from macrophages, which enhances T-cell-mediated cytotoxicity and increases NK cell infiltration within the tumor microenvironment [156]. Also, X. Wang et al. research presented a remarkable surge in PD-L1 expression on gastric cancer cell lines with the use of autophagy inhibitors chloroquine (CQ), 3-methyladenine, and bafilomycin A1 (Baf), which influenced the p62/SQSTM1-NF-κB pathway [157].

Based on the above understanding, it is obvious that HCQ, as a tool for autophagy inhibition, has contradictory outcomes. Owing to multiple factors, first is the ability of tumors to engage in alternative degradation pathways and compensatory survival mechanisms to bypass autophagy inhibition, such as the ubiquitin–proteasome system [152,158]. Second, with poor intratumoral penetration, HCQ accumulates in normal tissues (liver) rather than tumors [159]. Third, tumors that utilize the autophagy pathway to recycle cellular components and adapt to metabolic stress exhibit an increased sensitivity to autophagy inhibitors [159]. Fourth, in suboptimal and tolerable doses, researchers often use lower, safer doses, leading to incomplete autophagy inhibition and variable treatment responses, including immune suppressive effects [159]. In summary, this review underscored the complex interplay between immunotherapy and supportive treatments like autophagy modulators, revealing the potential pitfalls in their concurrent application.

**Table 4 cells-14-00745-t004:** Effects of autophagy modulation on CRC.

Tumor Types	Agent	Modulation	Related Mechanisms	Outcome	Reference
CRC	CXCL1	autophagy induction	reduce MHC-I expression	immune inhibition	[160]
CRC	Brucine	autophagy inhibition	enhance calreticulin and HMGB1 release	immune activation	[161]
CRC	FuFangChangTai Decoction	autophagy induction	activate macrophages and increase expression of CD86 and CD40	immune activation	[162]
CRC	Zosuquidar	PD-L1 selective autophagy	reduce PD-L1 expression	immune activation	[163]
CRC	Rigosertib	PD-L1 selective autophagy	reduce PD-L1 expression	immune activation	[164]
CRC	Nod1	Autophagy induction	M2 polarization	immune inhibition	[165]

CRC: colon rectal cancer, CXCL1: C-X-C motif chemokine ligand 1, HMGB1: high-mobility group box 1 protein.

**Table 5 cells-14-00745-t005:** Autophagy inhibitors enhance the efficacy of immunotherapy.

Target	Autophagy Inhibitor	Study Type	Clinical Trial Identifier	Outcome	References
PI3Kinase inhibition	Copanlisib + Nivolumab	Clinical Trial I/II	NCT03711058	Recruiting N.A.	[166]
Inhibition of VPS34	SB02024/SAR405	In vivo	-	Enhanced antitumor efficacy	[167]
Lysosomes, PPT1	Hydroxychloroquine + anti-PD-1	In vivo	-	Tumor growth impairment and improved survival in mouse models	[156]

PI3K: Phosphatidylinositol 3-kinase, VPS34: Vacuolar Protein Sorting 34, SB02024/SAR405: VPS34 inhibitors, PPT1: palmitoyl-protein thioesterase 1.

## 6. Conclusions and Perspectives

Enhancing the anticancer immune response using ICIs as anti-PD-1/PD-L1 is considered a valuable treatment option in CRC. However, the response to anti-PD-1/PD-L1 in such complex cancers as CRC requires certain genetic elements, such as the MMR gene expression. The status of MMR genes and tumor mutation burden serve as the main predictors of the immunotherapy response. However, the loss of expression of the MMR genes may not always indicate microsatellite instability status and hence immune responsiveness. The ultimate immune surveillance is guarded by two arms: first, CRC genetic makeup, including the location, quality of mutations, and phenotype of genes related to the immune system. Second, the TME, where an immunosuppressive environment engages in anti-PD-1/PDL-1 resistance throughout MDSC and Tregs that produce immunoinhibitory cytokines that downregulate the cytotoxic effect exerted by the immune system.

To overcome resistance and widen the therapeutic coverage of anti-PD-1/PD-L1, understanding the tumor microenvironment elements like autophagy and its very interesting roles in immunity, cytokines production, antigen presentation, and its interaction with different apoptotic pathways is considered a promising approach. This review sheds light on the complexity of the role of autophagy in immunity and the enhancement of its antitumor effect through many modulators in CRC. Autophagy is involved in antigen processing as it breaks down proteins into small peptide units to be recruited for MHCI/MHCII surface presentation, while inhibition of autophagy upregulates PD-L1 expression. PD-L1 expression, and presentation are crucial for T-cell activation and their cytolytic effect.

Combining autophagy modulation with anti-PD-1/PD-L1 therapy for CRC treatment is considered a novel area of research. The number of studies is limited and has resulted in two contradictory outcomes. The first supports autophagy inhibition as it has a favorable synergistic effect with anti-PD-1/PD-L1, while the second opinion reported the opposite. Both parties had reasonable findings, making it hard to decide whether up-/down-regulation of autophagy is favorable. This is understandable and predicted, as autophagy by itself can be a pro- and anti-tumor depending on the cancer stage and type. Also, CRC heterogenicity is governed not solely by autophagy but by other metabolic machinery that can compensate for nutrient deprivation when autophagy is absent. Moreover, autophagy is context-dependent and can convert CRC from being under immune surveillance to immune evasion or vice versa, based on its ability to deploy the immune checkpoint inhibitors’ surface presentation. Modulating autophagy at a specific time point during tumor development can significantly influence the result of such intervention. Also, the review highlights the main considerations related to the selection of optimum preclinical models that can simulate the mature functioning immune system to evaluate immunotherapies.

In addition, a proper preclinical model for testing autophagy was briefly mentioned to underline that choosing the preclinical model is critical for a better understanding of the autophagy process in various tumors, as it is integrated in many treatment modalities as a promising tool. This review recapitulates the relationship between autophagy’s dichotomous role in cancer immunity in CRC heterogenicity using anti-PD-1/PD-L1, introducing the main idea that autophagy overseeing can enhance the therapeutic outcomes of these agents via multiple pathways, which might serve as an encouraging mechanism. For the improvement of the anti-PD-1/PD-L1 clinical response in CRC immunotherapy-resistant subgroups, further studies are required to decipher the complex role of autophagy and its interaction with the immune system and other TME components to achieve optimal therapeutic outcomes. Also, tailoring such a combination according to candidates’ autophagic activity and PD-L1 expression can be beneficial.

## Figures and Tables

**Figure 1 cells-14-00745-f001:**
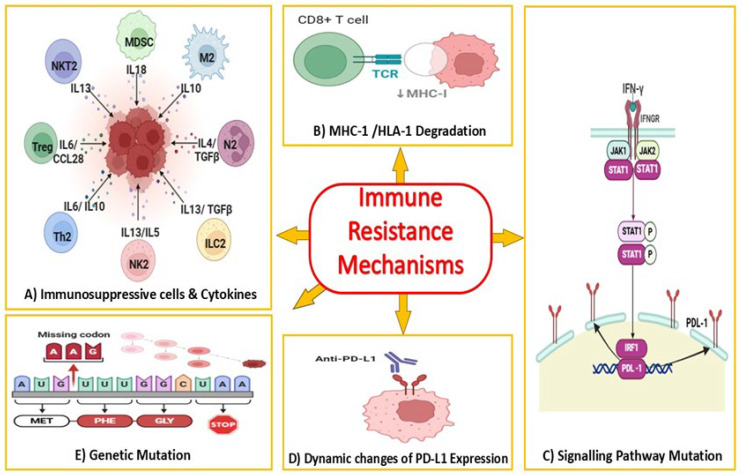
Immune Resistance Mechanisms. (**A**) Immunosuppressive Cells and Cytokines: Recruitment of Tregs, MDSCs, TAMs; secretion of IL-10, TGF-β, VEGF suppresses anti-tumor immune responses. (**B**) MHC/HLA-1 Degradation: Downregulation of MHC I; mutations in antigen-processing machinery, impaired antigen presentation, and immune recognition. (**C**) Signaling Pathway Mutations: JAK/STAT mutations: disrupt IFN-γ signaling, reducing PD-L1 upregulation or the immune-activating effects of interferons, promoting immune evasion and tumor survival. (**D**) Dynamic Changes of PD-L1 Expression: Tumors increase PD-L1 expression in response to inflammatory signals in the microenvironment. Tumors may downregulate PD-L1 expression temporarily to avoid detection by PD-1/PD-L1-targeting immunotherapies, enabling tumors to escape immune detection while adapting to changes in the immune microenvironment or therapy. (**E**) Genetic Mutations: Tumor cells acquire genetic mutations that directly or indirectly impact immune recognition and response. Drives immune evasion and resistance to immune checkpoint inhibitors. High mutational burden generates neoantigens that recruit the immune system, but certain mutations (e.g., in β2-microglobulin or JAK1/2) suppress immune recognition and response. NKT: natural killer T-cells, MDSC: myeloid-derived suppressor cells, TAMs: tumor associated macrophages, M2: macrophages, Th2: T helper cells, NK: natural killer cells, Tregs: regulatory T-cells, IL: interleukin, ILC2: type-2 innate lymphoid cells, MHC1: major histocompatibility complex, HLA: human leukocyte antigen, IFN-γ: interferon gamma, JAKs: Janus kinases, STAT: signal transducer and activator of transcription, IFNGR: interferon–gamma receptor, IRF1: interferon regulatory factor 1.

**Figure 2 cells-14-00745-f002:**
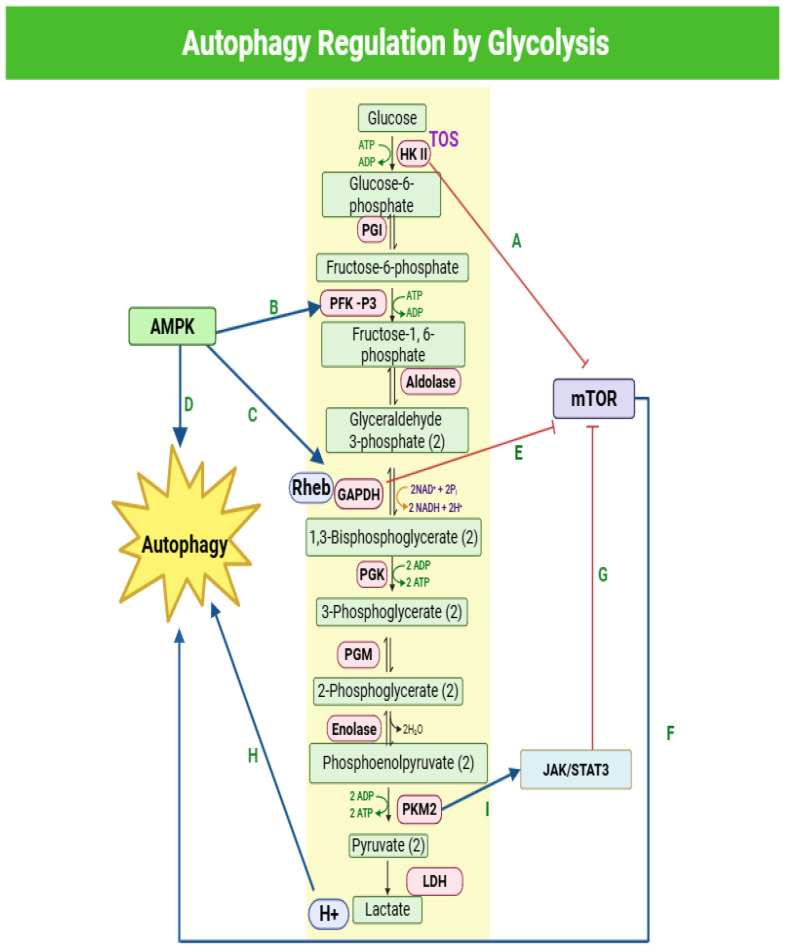
Autophagy regulation by glycolysis. A: The HK II protein contains a TOS motif, which allows it to inhibit mTOR by binding to it through this specific motif. As a result, it triggers the process of autophagy within the cell. B, C, D: During periods of starvation, AMPK is induced, which in turn stimulates the phosphorylation of two key glycolysis enzymes, PFKFB3 and GAPDH, and simultaneously promotes autophagy. E, F: GAPDH is known to form complexes with Rheb and has been found to inhibit the activity of mTOR; when mTOR is inhibited, it leads to the promotion of autophagy. G, H: PKM2 plays a role in promoting the JAK/STAT3 pathway and inhibiting the mTOR pathway, which ultimately leads to the promotion of autophagy. I: LDH is involved in the conversion of pyruvate to lactate while releasing protons as a byproduct. The released protons contribute to the acidification of autophagosomes, leading to an increase in acidity, which in turn promotes the process of autophagy. PGI: phosphoglucose isomerase, PGK: phosphoglycerate kinase, PGM: phosphoglycerate mutase, TOS: TOR signaling motif, HK II: Hexokinase 2, PFK-P3: phosphofructokinase, GAPDH: Glyceraldehyde 3-phosphate dehydrogenase, PKM2: pyruvate kinase M2, LDH: lactate dehydrogenase, AMPK: AMP-activated protein kinase, mTOR: mammalian target of rapamycin, Rheb: Ras homolog enriched in brain, STAT3: signal transducer and activator of transcription proteins, JAK: Janus kinase.

**Figure 3 cells-14-00745-f003:**
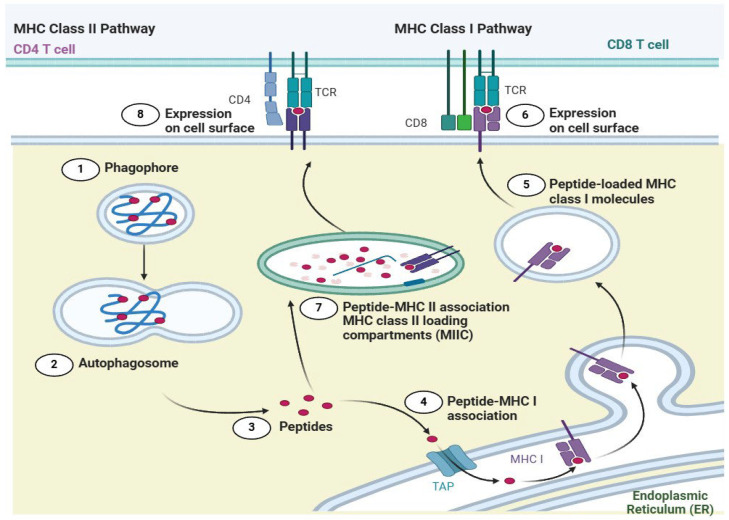
Antigen Processing and Presentation Via Autophagy. A multi-step process: (1) A double-membrane structure, called a phagophore, begins to form around cellular components (e.g., proteins, organelles, or pathogens). (2) The phagophore expands and engulfs intracellular materials to form a sealed structure called an autophagosome. (3) Degraded peptides from autophagy are transported into the endoplasmic reticulum (ER) via Transporter Associated with Antigen Processing (TAP) proteins. (4) Inside the ER, peptides are loaded onto MHC class I molecules. (5) Peptide-loaded MHC class II molecules are transported to the cell surface via the secretory pathway. (6) The antigens are displayed on the cell surface for recognition by cytotoxic T-cells (CD8+ T-cells). (7) The peptides generated during autophagy are transferred to MHC class II loading compartments (MIIC), in MIICs, MHC class II molecules bind these peptides. (8) On the cell surface, MHC class II molecules display the autophagy-derived peptides for recognition by helper T-cells (CD4+ T-cells).

**Figure 4 cells-14-00745-f004:**
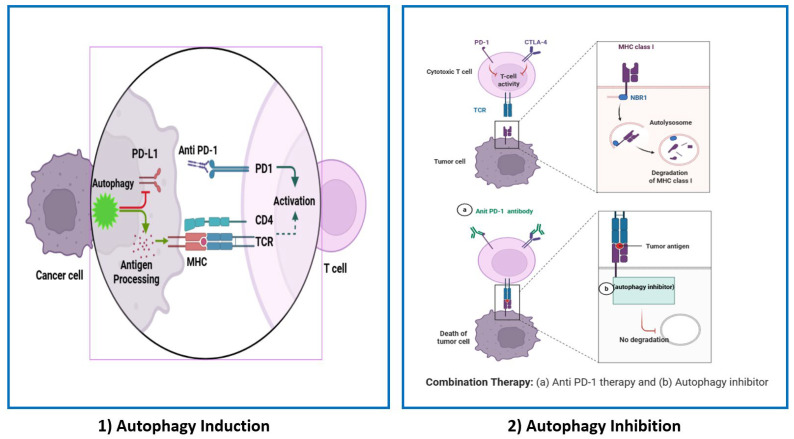
Targeting autophagy to enhance anti-PD-1/PD-L1 therapy. (1) Autophagy induction: facilitates the degradation of PD-L1 through lysosomal pathways; by facilitating this degradation, the signaling associated with immune checkpoints is diminished. Autophagy transports cytoplasmic antigens to lysosomes, where they are processed and then loaded onto MHC class II molecules. This process enhances the presentation of endogenous antigens to CD4+ T-cells. Additionally, autophagy can aid in the processing of antigens for presentation on MHC class I molecules, which influences CD8+ T-cell responses. MHC-I molecules are identified and bound by the protein NBR1. Following recognition, MHC-I components are transported to lysosomes, where they undergo degradation via the autophagic pathway. (2) Autophagy inhibition: impaired the degradation of PD-L1, leading to its accumulation on the cell surface and boosting the therapeutic efficacy of PD-L1 blockade.

## Data Availability

Not applicable.

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
