# Peer review of "Perspectives of Targeting Autophagy as an Adjuvant to Anti-PD-1/PD-L1 Therapy for Colorectal Cancer Treatment"

_cells, 2025, doi:10.3390/cells14100745_

Round 1
Reviewer 1 Report
Comments and Suggestions for Authors
The paper of Nasrah ALKhemeiri and co-authors summarizes current knowledge on anti-PD-1/PDL-1 treatment and the role of autophagy in cancer progression. Submitted paper corresponds to the current trends in cancer research as searching for new therapeutic targets in cancer is relevant topic in the field. The Authors reviewed the latest literature in the field, what should be emphasized as a great value of this article. Moreover it is clear that a lot of work has been done to summarize the knowledge and write this article. Unfortunately the manuscript generates too many concerns to be recommended for publishing without major revisions. Regardless of the scientific significance of this paper have raised some crucial comments:
Major comments:
- The lack of figures 1 and 4 makes it impossible to fully review the article.
- There are too many references cited in the article, the authors should rewrite the article and remove some of the references from the text. Especially from the fragments of the text when after one sentence there are more than two cited references, e.g. the first paragraph or lines: 407, 410, 574, 334, 345. The ratio between the number of text pages and reference pages is unusual.
- As far as too many references- why the same citation is put more than one time in the references section? See e.g. numbers 54, 221, 388. All citations should be checked.
- Some information could be considered for removing from the text. For example the general and known information about Warburg Effect, hydroxychloroquine or autophagy in the first paragraph of the section 3 ( the same info is given below the first paragraph) or in line 83, The removal will also help in reducing the number of cited works.
- The authors should reconsider the title of the article to put less impact on colorectal cancer in the title. The Authors focus on colorectal cancer, and, in fact, it is most described cancer in the text (mainly in the PD-1/PDL-1), but the main part of the manuscript concerns the aspects observed generally in many cancers (e.g. autophagy). Also the word “ combination” in the title is overestimated- Authors attempts to describe the combination between autophagy and anti-PD-1/PDL-1 treatment for CRC approaches are insufficient. The description in the manuscript is unclear. This should be clearly demonstrated, perhaps in a table, also showing the references?
- Also the title of paragraph 5. should be changed, because subtitle 5.2 relates to autophagy issues, and not to PD-1/PDL-1. Moreover, the first paragraph of 5 is too long and the sentences are too general.
- I also do not agree that “Autophagy as a key determinant modulator of immune response in colorectal cancer” (title of paragraph 3). First of all, autophagy is only one of modulator, but not the key one in cancer. Moreover, after reading the paragraph I am not convinced about the special role of autophagy in CRC in comparison to other cancers.
- Some sentences in the review are not easy to read and the meaning is difficult to understand, for example lines: 398
- I would be careful about referring to CRC in subtitles 3 and 3.1 because only a small part of the text refers to CRC. The same in the conclusions about CRC according to the aim of the review (line 681).
- In the lines 426 and 429: the authors claim that they summarize autophagy effect as far as CRC. I think such conclusions are overestimated, because the papers cited above refer to different cancers (e.g. glioblastoma). I suggest to rewrite the paragraph started from the line 429 with precise indication of publications relating to CRC.
- The authors must standardize abbreviations used in the text as far as PD-1/PDL-1 (e.g. see lines: 528, 568) and as far as MSS/pMMR and MSI-H/dMMR (authors sometimes use half of the name (as 221, 222), sometimes they change the order between the slash), see also MMRp vs. pMMR. The Authors should use all names consistently in the whole text and the tables.
- In the title of table 3 and the title of the first column in the table and in line 201 “MMS” is used instead of MSS.
- The authors should pay more attention to the table 1. It should be corrected and transform in some extent. The first column should have different title, because the authors show not only CRC below. This column should be divided according to Clinical Trail Identifier, because it should be precisely shown which cancer is taken into consideration in each trial. Also the authors should review table 2, where for example, there is NCT03168139- and it is not written that the trial also applies to pancreatic cancer.
- Some of the sentences are incomprehensible e.g. line 67and 78 – the meaning of words “these agents”.
- The authors have forgotten to include cancer cells into TME- line 289.
- I cannot agree that Phosphatidylinositol 3-kinase is the same as VSP34.
Minor comments:
- In line 81- there is logical error in the sentence. PD-1/PDL-1 were approved by the FDA because they have shown the most promising treatment outcome to date. The authors should mind cause and effect relationship.
- Line 94 – instead of “ instability” – “stability level” is more correct
- Line 144- MSI-L appears, how does it relate to MSS or MSI-H? The role of a review authors is to transform and combine descriptions from different publications.
- why “ MSI-H” is written in the table 2 legend?
- Line 112: mCRC used for the first time- abbreviation should be explained
- Line 80: “Anti-programmed death-1” should be changed to the correct form
- There are a lot unnecessary spaces, e. g. lines: 99, 122, 137, 169, 207, 281, 263, 600 and others
- Please, mind: unecessary capital letters in the sentence as in line 199, 282, 382, brackets as in 225 or author’s inicials as in 430, “7” in 501
Comments on the Quality of English Language
Some sentences in the text are not easy to read and the meaning is difficult to understand, because of they complexity.
Author Response
We thank the reviewer for the insightful comments and valuable suggestions that helped us markedly improve the manuscript. Kindly find below the authors’ response to your comments one by one.
Comment 1: The lack of figures 1 and 4 makes it impossible to fully review the article.
Response 1: figures 1 and 4 are attached in attachments section. figure 1 is cited in line 242 and figure 4 in line 578.
Comment 2: There are too many references cited in the article, the authors should rewrite the article and remove some of the references from the text. Especially from the fragments of the text, when after one sentence there are more than two cited references, e.g. the first paragraph or lines: 407, 410, 574, 334, 345. The ratio between the number of text pages and reference pages is unusual.
Response 2: Thank you for your comment. The references were revised and kept as minimum as possible, without overlooking important ones.
Comment 3: As far as too many references- why the same citation is put more than one time in the references section? See e.g. numbers 54, 221, 388. All citations should be checked.
Response 3: Thank you, all the citations have been checked to avoid duplication.
Comment 4: Some information could be considered for removing from the text. For example, the general and known information about Warburg Effect, hydroxychloroquine or autophagy in the first paragraph of the section 3 (the same info is given below the first paragraph) or in line 83, The removal will also help in reducing the number of cited works.
Response 4: Thank you, the number of cited references is reduced. Paragraph 3 is introductory to autophagy in general (definition, types, roles). In paragraph 3.1 the mention of Warburg Effect and Glycolysis is key to highlight the role of autophagy in CRC microenvironment, keeping it informative yet much shorter than the first version. line 83 will be kept as it addresses PD-1/PDL-1.
Comment 5: The authors should reconsider the title of the article to put less impact on colorectal cancer in the title. The Authors focus on colorectal cancer, and, in fact, it is most described cancer in the text (mainly in the PD-1/PDL-1), but the main part of the manuscript concerns the aspects observed generally in many cancers (e.g. autophagy). Also, the word “combination” in the title is overestimated- Authors attempts to describe the combination between autophagy and anti-PD-1/PDL-1 treatment for CRC approaches is insufficient. The description in the manuscript is unclear. This should be clearly demonstrated, perhaps in a table, also showing the references?
Response 5: The title is reconsidered and the word “combination” is removed. Regarding the combination of autophagy modulators and anti-PD-1/PDL-1 treatment for CRC and other types of cancer, this approach is evolving, with promising therapeutic outcomes. The number of studies reporting such a combination is limited, yet justified for many reasons. to achieve the optimum desired outcome, we need to have a reasonable autophagy modulator. For example, the widely used FDA approved autophagy inhibitor; HCQ and CQ have been restricted by their bioavailability, selectively and their off-target effects. So, all the reports are more toward finding new modulators to achieve optimal autophagy modulation to serve the objective of enhancing the anti-PD-1/PD-L1 treatment. We tried to list the available studies in this context as we want to attract more attention to the value of autophagy modulations to enhance the efficacy of anti-PD-1/PD-L1 in solid tumors, especially the CRC. Keeping in mind that 85% of CRC cases are not susceptible to anti-PD-1/PD-L1 treatment, adding to the fact that the treatment option for CRC are confined to surgery and chemotherapy, including the anti-PD-1/PD-L1 treatment could be a valuable addition to CRC therapeutics options.
Comment 6: Also, the title of paragraph 5. should be changed, because subtitle 5.2 relates to autophagy issues, and not to PD-1/PDL-1. Moreover, the first paragraph of 5 is too long, and the sentences are too general.
Response 6: We changed the paragraph title according to your valuable comment. we added autophagy in the title, and the paragraph was modified based on your recommendation to make it clearer and more understandable and to serve the purpose that we intend to highlight.
Comment 7: I also do not agree that “Autophagy as a key determinant modulator of immune response in colorectal cancer” (title of paragraph 3). First of all, autophagy is only one of modulator, but not the key one in cancer. Moreover, after reading the paragraph I am not convinced about the special role of autophagy in CRC in comparison to other cancers.
Response 7: Thanks for the valid point. I agree that there are other modulators (mentioned in the text), so autophagy can’t be considered as a key determinant. The words “key determinant” are removed from the title of paragraph 3. Moreover, paragraph 3 is an overview of the subject, and the following paragraphs will be more detailed.
Comment 8: Some sentences in the review are not easy to read and the meaning is difficult to understand, for example lines: 398
Response 8: The article was thoroughly revised and modified for better comprehension.
Comment 9: I would be careful about referring to CRC in subtitles 3 and 3.1 because only a small part of the text refers to CRC. The same in the conclusions about CRC according to the aim of the review (line 681).
Response 9: Agreed as paragraph 3 and 3.1 address autophagy general role in both cancer and its microenvironment, therefore, the titles of paragraph 3 and 3.1 are changed to represent the content.
Comment 10: In the lines 426 and 429: The authors claim that they summarize autophagy effect as far as CRC. I think such conclusions are overestimated, because the papers cited above refer to different cancers (e.g. glioblastoma). I suggest to rewrite the paragraph started from the line 429 with precise indication of publications relating to CRC.
Response 10: We removed the reference (glioblastoma). We added 2 extra tables (Table 4 and Table 5) to highlight the autophagy in CRC only, and not for other cancers.
Comment 11: The authors must standardize abbreviations used in the text as far as PD-1/PDL-1 (e.g. see lines: 528, 568) and as far as MSS/pMMR and MSI-H/dMMR (authors sometimes use half of the name (as 221, 222), sometimes they change the order between the slash), see also MMRp vs. pMMR. The Authors should use all names consistently in the whole text and the tables.
Response 11: Thank you, I standardize all the abbreviations.
Comment 12: In the title of table 3 and the title of the first column in the table and in line 201 “MMS” is used instead of MSS.
Response 12: corrected as recommended.
Comment 13: The authors should pay more attention to the table 1. It should be corrected and transform in some extent. The first column should have different title, because the authors show not only CRC below. This column should be divided according to Clinical Trial Identifier, because it should be precisely shown which cancer is taken into consideration in each trial. Also the authors should review table 2, where for example, there is NCT03168139- and it is not written that the trial also applies to pancreatic cancer.
Response 13: The table is rearranged as suggested, and NCT03168139 clinical trial is added to table2. With regards to the first column title of table 1, all the listed clinical trials recruited patients with CRC and other types of cancers which we mentioned to describe the clinical trials.
Comment 14: Some of the sentences are incomprehensible e.g. line 67and 78 – the meaning of words “these agents”.
Response 14: changed as suggested in line 67 “these agents” changed to anti-PD-1/PDL-1, line 78 “these agents” changed to ICIs.
Comment 15: The authors have forgotten to include cancer cells into TME- line 289.
Response 15: We followed the definition of tumor microenvironment according to the National Cancer Institute: it is "the normal cells, molecules, and blood vessels that surround and feed a tumor cell".
https://www.cancer.gov/publications/dictionaries/cancer-terms/def/microenvironment
Minor comments:
Comment 1: In line 81- there is logical error in the sentence. PD-1/PDL-1 were approved by the FDA because they have shown the most promising treatment outcome to date. The authors should mind cause and effect relationship.
Response 1: edited as recommended
Comment 2: Line 94 – instead of “ instability” – “stability level” is more correct.
Response 2: edited as recommended
Comment 3: Line 144- MSI-L appears, how does it relate to MSS or MSI-H? The role of a review authors is to transform and combine descriptions from different publications.
Response 3: edited as recommended. MSI-L is the same as MSS/pMMR.
Comment 4: why “ MSI-H” is written in the table 2 legend?
Response 4: MSI-H is mentioned ( Non- MSI-H mCRC) in row number 10 of table 2, therefore, it is added to the legend.
Comment 5: Line 112: mCRC used for the first time- the abbreviation should be explained
Response 5: explained as recommended.
Comment 6: Line 80: “Anti-programmed death-1” should be changed to the correct form
Response 6: corrected to Anti-programmed cell death protein 1, as recommended.
Comment 7: There are a lot unnecessary spaces, e. g. lines: 99, 122, 137, 169, 207, 281, 263, 600 and others
Response 7: removed all the unnecessary spaces.
Comment 8: Please, mind: unnecessary capital letters in the sentence as in line 199, 282, 382, brackets as in 225 or author’s initials as in 430, “7” in 501
Response 8: Edited as recommended. Unless it is drug names, disease or created by citation tool.
Reviewer 2 Report
Comments and Suggestions for Authors
The review article is related to the role of autophagy with respect to immunotherapy by immune checkpoint inhibition (ICI) in colorectal cancer, thus the topic is quite unusual.
It should be mentioned early in the text that ICI so far is only approved and clinically effective in patients with defects in mismatch repair (dMMR).
The role of tumor mutational burden should be mentioned and explained in the context of dMMR/MSI high together with respect to the activity of ICI therapy in this situation.
The tables 1 and 2 are not very informative and should be improved: the most important trial should be included and more outcome information as well as the citation should be included. In Table 2 important aspects are lacking for example the data on Regorafinib/Nivolumab in pMMR colorectal cancer patients with no liver metastasis (Fakih M, et al. Regorafenib plus nivolumab in patients with mismatch repair-proficient/microsatellite stable metastatic colorectal cancer: a single-arm, open-label, multicentre phase 2 study. EClinicalMedicine. 2023;58:101917. doi:10.1016/j.eclinm.2023.101917.
Overall, this review article is not very coherent. Different aspects are described (colorectal cancer biology, the role of immune checkpoint therapy in colorectal cancer, the role of autophagy on antigen presentationetc.), however the overarching rational and data that would clearly demonstrate that autophagy is playing an important role in colorectal cancer treatment with immune checkpoint inhibitors are lacking. Only Zhang et al. as cited provides a link between an autophagy related protein (AGT7) and immune checkpoint blockade in MSI high colorectal cancer from TCGA data. The other data cited in the review are just in vitro studies. In summary, although all aspects mentioned in the title are covered in this review article, the database to support the link of autophagy, immune checkpoint therapy and colorectal cancer, especially pMMR is extremely small and from this background it can not very well be understood, why an extensive review in the end is based on such a small data basis regarding the key topic as indicated in the title.
Line 253: it would be important to note that is metastatic colorectal cancer the percentage of MSS tumors is as high as 95%.
The term “immunogenic cell death is used (line 334) however, no explanation is given, what this means.
Minor remarks: Line 60 PD-L1 instead of PDL-1 to be corrected in the entire text. Abreviation MMS is not explained (microsatellite stable? MS? Why MMS?). In other part of the manuscript MSI-L. Please use a common terminology and abbreviations.
Comments on the Quality of English Language
none
Author Response
The authors would like to thank the reviewer for the insightful comments and valuable suggestions that helped us markedly improve the manuscript. Kindly find below the authors’ response to your comments one by one.
Comment 1: The review article is related to the role of autophagy with respect to immunotherapy by immune checkpoint inhibition (ICI) in colorectal cancer, thus, the topic is quite unusual.
Response 1: The topic can be considered new with a few clinical data, yet of promising prospects that may guide the proper selection of appropriate candidate patients, diagnosed with CRC of types other than the MSI-high ones.
Comment 2: It should be mentioned early in the text that ICI so far is only approved and clinically effective in patients with defects in mismatch repair (dMMR).
Response 2: This statement is mentioned and explained in paragraph 2.1 line 99 to 101.
Comment 3: The role of tumor mutational burden should be mentioned and explained in the context of dMMR/MSI high together with respect to the activity of ICI therapy in this situation.
Response 3: mentioned in paragraph 2.1 line 95 to 98 and explained in figure 1.
Comment 4: The tables 1 and 2 are not very informative and should be improved: the most important trial should be included and more outcome information as well as the citation should be included. In Table 2 important aspects are lacking for example the data on Regorafinib/Nivolumab in pMMR colorectal cancer patients with no liver metastasis (Fakih M, et al. Regorafenib plus nivolumab in patients with mismatch repair-proficient/microsatellite stable metastatic colorectal cancer: a single-arm, open-label, multicentre phase 2 study. EClinicalMedicine. 2023;58:101917. doi:10.1016/j.eclinm.2023.101917.
Response 4: Tables 1 and 2 have been edited and citations have been added. the data on Regorafinib/Nivolumab in pMMR colorectal cancer patients with no liver metastasis (Fakih M, et al. Regorafenib plus nivolumab in patients with mismatch repair-proficient/microsatellite stable metastatic colorectal cancer: a single-arm, open-label, multicentre phase 2 study. EClinicalMedicine. 2023;58:101917. doi:10.1016/j.eclinm.2023.101917), is also added to table 2.
Comment 5: Overall, this review article is not very coherent. Different aspects are described (colorectal cancer biology, the role of immune checkpoint therapy in colorectal cancer, the role of autophagy on antigen presentation etc.), however the overarching rational and data that would clearly demonstrate that autophagy is playing an important role in colorectal cancer treatment with immune checkpoint inhibitors are lacking. Only Zhang et al., as cited, provides a link between an autophagy related protein (AGT7) and immune checkpoint blockade in MSI high colorectal cancer from TCGA data. The other data cited in the review are just in vitro studies. In summary, although all aspects mentioned in the title are covered in this review article, the database to support the link of autophagy, immune checkpoint therapy and colorectal cancer, especially pMMR is extremely small and from this background it can not very well be understood, why an extensive review in the end is based on such a small data basis regarding the key topic as indicated in the title.
Response 5: Thanks for your valuable comments. We discussed the aspects mentioned above to introduce the topic and give detailed information about the many roles of autophagy in cancer, in general and in CRC specifically. Moreover, in paragraph 3.3 many citations highlight the relation between autophagy proteins and cancer. Table 3 shows how autophagy leads to drug-resistance. Table 4 shows how autophagy modulation affects CRC. Table 5 shows the combine effect of autophagy inhibition +anti-PD-1 on CRC. I agree that the information on autophagy about ICI in CRC is very limited due to little research on it, but I tried to overcome that by mentioning examples from other cancers, still relevant to extrapolate in case of CRC, due to some similarities.
Comment 6: Line 253: it would be important to note that is metastatic colorectal cancer the percentage of MSS tumors is as high as 95%.
Response 6: This information is added in line 104 as it is more relevant there.
Comment 7: The term “immunogenic cell death is used (line 334) however, no explanation is given, what this means.
Response 7: Immunogenic cell death is explained in paragraph 3.1 lines 343 to 347.
Minor remarks:
Comment 1: Line 60 PD-L1 instead of PDL-1 to be corrected in the entire text. abbreviation MMS is not explained (microsatellite stable? MS? Why MMS?). In other part of the manuscript MSI-L. Please use a common terminology and abbreviations.
Response 1: I agree that PD-L1 is the correct term, and it is corrected to the whole text. We standardized all the abbreviations.
Reviewer 3 Report
Comments and Suggestions for Authors
The aim of this review article is to summarize the potential of PD-1/PD-L1 ICIs and autophagy inhibitors in combination for CRC. The topic is very intriguing and important, both for basic research and for clinical work and therapy development.
However, the following fundamental shortcomings in the approach are present in the article:
Chapter 3.0 does not discuss the role of ULK1-mediated phosphorylation inhibitors or ULK1 complex inhibitors as possible ways to block autophagy. It also does not mention the phagophore nucleation inhibitor VPS34, the vacuolar ATPase inhibitors, or the palmitoyl-protein thioesterase 1 inhibitors.
The section on the combined use of autophagy and ICI inhibitors (4.0) is the shortest in the article, even though it is the main objective. This chapter, which contains important information, is out of balance with the other chapters, which summarize the facts that are already well known.
Didactically, chapter 5.0 would be logical before chapter 4.0. Chapter 5.0 describes separately the animal models for studying the inhibition of PD1/PDL1 and autophagy but does not synthesize how it would be useful to study the co-modulation of the two pathways.
L49-52: Small molecule inhibitors and biological treatment must be mentioned.
A major English language polishing is necessary.
Comments on the Quality of English LanguageSeveral typos, grammatical and synthactical mistakes can be detecetd in the text. A major language polishing is required.
Author Response
We thank the reviewer for the insightful comments and valuable suggestions that helped us markedly improve the manuscript. Kindly find below the authors’ response to your comments one by one.
Comment 1: Chapter 3.0 does not discuss the role of ULK1-mediated phosphorylation inhibitors or ULK1 complex inhibitors as possible ways to block autophagy. It also does not mention the phagophore nucleation inhibitor VPS34, the vacuolar ATPase inhibitors, or the palmitoyl-protein thioesterase 1 inhibitors.
Response 1: ULK1 role in autophagy is explained in paragraph 3.1 lines 312 to 316. VPS34 is mentioned in paragraph 3.2 lines 386 to 390. palmitoyl-protein thioesterase 1 mentioned in paragraph 5 lines 645 to 651.
Comment 2: The section on the combined use of autophagy and ICI inhibitors (4.0) is the shortest in the article, even though it is the main objective. This chapter, which contains important information, is out of balance with the other chapters, which summarize the facts that are already well known.
Response 2: Kindly note that lines 520 to 572 is also addressing the combination therapy and it directly follows paragraph 4. Therefore, paragraph 4 plus lines 520 to 572 are collectively well balanced with other paragraphs.
Comment 3: Didactically, chapter 5.0 would be logical before chapter 4.0. Chapter 5.0 describes separately the animal models for studying the inhibition of PD1/PDL1 and autophagy, but does not synthesize how it would be useful to study the co-modulation of the two pathways.
Response 3: We resolved this issue based on your valid suggestion. Chapter 5 was moved to become chapter 4, and we added a paragraph on the combination. We modified the paragraph to be more focused, and to highlight the importance of preclinical model selection in the study of immunotherapy and for monitoring autophagy. I mentioned in the same paragraph that the number of studies is not as much as this area is new and needs to be investigated deeply to achieve the desired therapeutic outcome.
Comment 4: L49-52: Small molecule inhibitors and biological treatment must be mentioned.
Response 4: Done, “small molecule inhibitors” was added.
Comment 5: A major English language polishing is necessary.
Response 5: Done. The review was thoroughly revised. The spelling and grammar were meticulously checked.
Round 2
Reviewer 2 Report
Comments and Suggestions for Authors
All points have been very well addressed and with the changes made, the manuscript is now ready for publication.
Reviewer 3 Report
Comments and Suggestions for Authors
I accept the answers and explanations. The revision was carried out according to the suggestions.